# The E-Cadherin and N-Cadherin Switch in Epithelial-to-Mesenchymal Transition: Signaling, Therapeutic Implications, and Challenges

**DOI:** 10.3390/cells8101118

**Published:** 2019-09-20

**Authors:** Chin-Yap Loh, Jian Yi Chai, Ting Fang Tang, Won Fen Wong, Gautam Sethi, Muthu Kumaraswamy Shanmugam, Pei Pei Chong, Chung Yeng Looi

**Affiliations:** 1School of Biosciences, Faculty of Health and Medical Sciences, Taylor’s University, Subang Jaya 47500, Malaysia; cyloh5@gmail.com (C.-Y.L.); benjaminchaijy@gmail.com (J.Y.C.); peipei.chong@taylors.edu.my (P.P.C.); 2Department of Medical Microbiology, Faculty of Medicine, University of Malaya, Kuala Lumpur 50603, Malaysia; tiffanytftang@gmail.com; 3Department of Pharmacology, Yong Loo Lin School of Medicine, National University of Singapore, Singapore 117597, Singapore

**Keywords:** E-cadherin, N-cadherin, Epithelial-to-Mesenchymal Transition, signaling pathways, natural compounds

## Abstract

Epithelial-to-Mesenchymal Transition (EMT) has been shown to be crucial in tumorigenesis where the EMT program enhances metastasis, chemoresistance and tumor stemness. Due to its emerging role as a pivotal driver of tumorigenesis, targeting EMT is of great therapeutic interest in counteracting metastasis and chemoresistance in cancer patients. The hallmark of EMT is the upregulation of N-cadherin followed by the downregulation of E-cadherin, and this process is regulated by a complex network of signaling pathways and transcription factors. In this review, we summarized the recent understanding of the roles of E- and N-cadherins in cancer invasion and metastasis as well as the crosstalk with other signaling pathways involved in EMT. We also highlighted a few natural compounds with potential anti-EMT property and outlined the future directions in the development of novel intervention in human cancer treatments. We have reviewed 287 published papers related to this topic and identified some of the challenges faced in translating the discovery work from bench to bedside.

## 1. Introduction

Cadherins were originally identified by Takeichi as cell surface molecules involved in Ca^2+^-dependent adhesion mechanism in Chinese hamster V79 cells [1]. Cadherins are highly sensitive to Ca^2+^ and are readily degraded by proteolysis in the absence of Ca^2+^ [2]. The cadherin superfamily consists of 114 calcium-dependent membrane proteins from three major cadherin families (cadherins, protocadherins and cadherin-related proteins), and these proteins from different families differ in many aspects from each other [3,4]. Different cadherins share similar overall primary structures, with their mature forms consisting of 723 to 748 amino acids [2]. Cadherins are involved in regulating cell–cell adhesion as well as modulating crucial morphogenetic and differentiation processes during development [5,6]. 

The classical cadherins are single span transmembrane cadherins with five extracellular cadherin (EC) repeat domains that cooperate with catenin family members via their cytoplasmic domains to link with the underlying actin cytoskeleton [7,8]. Classical cadherins are grouped into type-I and type-II subgroups based on the molecular features of their interaction via the cadherin motif [7]. Classical cadherins mediate segregation of different cell populations based on homophilic adhesion mechanism where the cells expressing different cadherin subtypes keep apart from each other [9]. A “strand-dimer” model has been widely accepted for the homophilic adhesion mechanism of type I cadherins [10]. This model is a trans interaction involving six highly conserved N-terminal residues of EC1 known as “adhesion arm” and the conserved “acceptor pocket” in the body of EC1, in which the adhesion arm from one cadherin inserts into acceptor pocket of the opposing cadherin [10]. 

E- and N-cadherin belong to type-I classical cadherins alongside P-cadherin, R-cadherin and M-cadherin. The size of mature E-cadherin protein is 120 kDa while mature N-cadherin protein is 130 kDa. Structure of E- and N-cadherin are as illustrated in Figure 1. Cadherin cytoplasmic tails attach with β-catenin which connects to α-catenin, thus form the cadherin-catenin adhesion complex [11]. Although α-catenin and β-catenin are crucial in the binding of cadherin to F-actin, they cannot bind F-actin directly; hence, additional adaptors such as vinculin, eplin or zona occludens protein 1 (ZO1) are required (Figure 1) [12]. Interestingly, in comparison with the E-cadherin adhesions, the wild-type vinculin is highly extended in N-cadherin adhesions [13]. The expression of α-catenin determines the z-positioning of vinculin and emplaces vinculin in the interface zone, allowing it to bridge the cadherin-catenin and actin compartments [13]. This concurrently regulates the position of zyxin and vasodilator-stimulated phosphoprotein (VASP), which in turn regulates actin [13]. Indeed, a recent study demonstrated that the force-dependent vinculin/α-catenin association is vitally important in the formation of adaptive cell–cell contacts in response to mechanical stress, long-range cell–cell interactions and tissue-scale mechanics [11]. Previous studies showed that the anchoring of cadherin complexes to F-actin mediated by vinculin can restore E-cadherin-dependent cell contacts without the presence of α-catenin actin binding domain [14]. The cadherin–catenin adhesion is the essential component of adherens junctions that maintain tissue stability and dynamic cell movements [11]. Adherens junctions are regulated at the levels of transcription, translation, and trafficking, as well as via post-translational regulation of cadhesome proteins including serine or threonine phosphorylation, tyrosine phosphorylation, and proteolysis [12]. Tension also plays a critical role in the maturation process of cadherin-mediated adhesions. 

E-cadherin is one of the well-studied founding member of the superfamily and a potent tumor suppressor because downregulation of E-cadherin is often found in malignant epithelial cancers [15,16,17]. E-cadherin is important in maintaining the epithelial phenotype and regulating homeostasis of tissues by modulating various signaling pathways [18]. Mutations, proteolytic cleavage, chromosomal deletions, epigenetic regulation and transcriptional silencing of CDH1 promoter were shown to impede the functionality of E-cadherin in the development of several malignancies including gastric, breast, liver, pancreas, and skin [19,20,21,22,23].

Loss of E-cadherin in cancer cells leads to metastatic dissemination and activation of several EMT transcription factors [24]. Ovarian cancer cell lines with higher E-cadherin expression showed poorer resistance to cell death, lower adhesion to extracellular matrices and weaker invasiveness in comparison to cell lines with higher N-cadherin expression [17]. E-cadherin is proven to promote nucleotide excision repair by increasing the expression of xeroderma pigmentosum complementation group C (XPC) and DNA damage-binding protein 1 (DDB1) in the event of ultraviolet-induced DNA damage [25]. 

Uncontrolled growth due to dysfunctional contact inhibition of proliferation is one of the specific features of solid cancers [26,27]. Homophilic binding of E-cadherin inhibits cell proliferation by regulating expression levels of receptor tyrosine kinase (RTK) and tyrosine kinase Src in the absence of cell–cell interaction [28]. E-cadherin-expressing cell line with elevated nuclear factor kappa-light-chain-enhancer of activated B cells (NF-κB) activity and upregulated c-Myc expression was demonstrated to promote cell proliferation following the increase in adenosine triphosphate (ATP) production by glycolysis and mitochondrial oxidative phosphorylation (OXPHOS) [29]. Cancer cells commonly use aerobic glycolysis or mitochondrial OXPHOS, depending on the circumstances, to acquire ATP as energy to sustain their high rates of proliferation [30,31,32]. 

Most studies suggested that decrease of E-cadherin expression is the hallmark of EMT, but some researchers argued that the loss of E-cadherin is not causal nor a necessity for EMT and restoration of E-cadherin expression in E-cadherin negative malignant cells did not reverse the EMT [33]. Nilsson et al. also demonstrated that E-cadherin loss is consequential rather than causal for c-erbB2-induced EMT in non-malignant mammary epithelial cell line HB2 [34]. Loss of E-cadherin alone has been proved to be insufficient to drive EMT program in non-malignant breast cell line MCF10A [35]. However, the loss of E-cadherin expression has always been associated with more aggressive and less differentiated malignant cells. Besides, the loss of E-cadherin has been reported to cause activation of several EMT transcription factors, as aforementioned. Therefore, the loss of E-cadherin is a major driver or consequence of EMT remains a question to be answered. 

Various invasive and metastatic cancers have been associated with high level of E-cadherin expression, notably in patients with prostate cancer [36], ovarian cancer [37], and glioblastoma [38], thus suggesting that E-cadherin promotes metastasis in certain tumors instead of suppressing tumor progression. Dual roles of E-cadherin is possibly due to the existence of two forms of E-cadherins which are membrane-tethered E-cadherin as aforementioned and soluble E-cadherin (80 kDa) [39]. When the membrane-tethered E-cadherin is cleaved, soluble E-cadherin containing five repeated domains at the N-terminal will be released. Soluble E-cadherin was initially discovered by Wheelock et al. [40] in the conditioned media of MCF-7 cells and subsequent studies have been carried out to investigate the suitability of soluble E-cadherin as a cancer biomarker [41]. For example, soluble E-cadherin has been employed as one of the markers of EMT and tumor invasion in a phase I clinical trial in non-small cell lung cancer (NSCLC) patients [42], phase II trial in gastric cancer patients [43], and phase II trial in metastatic renal cell carcinoma patients [44]. Soluble E-cadherin accumulation interferes with adherens junctions, increases cancer cell migration, proliferation, and survival by promoting the activity of a disintegrin and metalloprotease (ADAM) and matrix metalloproteinases (MMPs) as well as manipulating several signaling pathways [39,45]. Recently, Tang et al. demonstrated that soluble E-cadherin is highly expressed in ovarian cancer patients where soluble E-cadherin is released in the form of exosomes to promote tumor angiogenesis via activation of β-catenin and NF-κB signaling [45].

In contrast with E-cadherin, N-cadherin is prevalent in non-epithelial tissues and is expressed in different types of cells such as neural cells, endothelial cells, stromal cells, and osteoblasts [46,47]. In neural tissue, N-cadherin replaces E-cadherin during neurulation and forms strong adherens junctions to maintain the tissue architecture of neural tissues as well as regulate proliferation and differentiation of neural progenitor cells [48]. N-cadherin serves as an indicator of ongoing EMT and its expression has been correlated with the development of various types of carcinoma [49,50,51,52]. N-cadherin promotes angiogenesis and maintains the integrity of blood vessels by ensheathing endothelial cells and mural cells for stabilization of microvessels [53,54,55]. Loss of N-cadherin leads to an increase of pancreatic intraepithelial neoplasia and tumor incidence in oncogenic K-ras^G12D^ model, suggesting that it could function as a growth suppressor [56]. A similar role was reported in neuroblastoma, where downregulation of N-cadherin leads to metastatic dissemination [57]. 

In short, E-cadherin-mediated adherens junctions suppress activation of Wnt/β-catenin pathway and RTK/P13K pathway in epithelial cells (Figure 2). N-cadherin-mediated adherens junctions facilitate the activation of mitogen-activated protein kinase (MAPK)/extracellular signal-regulated kinases (ERK) and Phosphoinositide-3-kinase (PI3K) pathways in association with platelet-derived growth factor receptor (PDGFR) to enhance cell survival and migration in non-epithelial cells (Figure 2). The functions of E- and N-cadherin are summarized in Figure 2 and the signaling pathways are further discussed in the following section.

## 2. E-Cadherin and N-Cadherin in Epithelial-to-Mesenchymal Transition 

Epithelial cells could go through EMT to acquire mesenchymal characteristics following the dissociation of cell–cell junctions and suppression of E-cadherin [58,59]. Dissociation of cell–cell junctions including tight junction, gap junction, adherens junction and desmosome leads to the loss of epithelial integrity [59]. The adherens junctions are comprised of cadherins that are associated with the actin cytoskeleton via binding to α-, β-, and γ-catenins [60]. During EMT, cleavage of E-cadherin leads to the destabilization of adherens junctions and the release of β-catenin to function as a transcriptional activator for cell proliferation [61,62]. EMT and its opposite mesenchymal-to-epithelial transition (MET) are not binary processes, cancer cells are reported to undergo partial EMT and exhibit a hybrid epithelial/mesenchymal phenotype, which can be more tumorigenic and chemoresistance in comparison to cancer cells that have predominant epithelial or mesenchymal features [63,64]. 

Ras Homolog Gene Family member A (RhoA) and RAC are GTPases that form mutually repressing feedback loops and play important roles during EMT in reversing the apico-basal polarity of epithelial cells to a front–back polarity (Figure 3) [64]. RAC especially Ras-related C3 Botulinum Toxin Substrate 1 (Rac1) induces cytoskeletal reorganization in the cells to enhance their directional migration prowess while RhoA induces the formation of actin stress fibers and the stabilization of N-cadherin mediated adherens junctions [64].

N-cadherin is upregulated while E-cadherin is downregulated during EMT in cancers and this “cadherin switch” is associated with enhanced migratory and invasive traits, which caused inferior patient survival rate [47,65,66]. Activation of some notable pathways and EMT program by the “cadherin switch” are summarized in Figure 3.

N-cadherin induces EMT and cancer stem cell–like characteristics by activating ErbB pathway through upregulating the level of growth factor receptor-bound protein 2 (GRB2), SHC-transforming protein and ERK [49]. An increase of pluripotency-associated markers was observed in prostate cancer cells that were overexpressing N-cadherin [49]. N-cadherin facilitates tumor–host contacts with N-cadherin expressing cells to enhance tumor cell migration and proliferation [51,67]. A study exhibited that N-cadherin activates MAPK/ERK pathway in polyomavirus middle T antigen (PyVmT)-infected mice [51]. However, N-cadherin can inhibit the EMT and establish strong adherens junctions in the neural crest cells [68]. 

In cancer, EMT selectively inhibits apoptosis signaling via the death receptor 4 TNF-related apoptosis-inducing ligand-receptor 1 (TRAIL-R1) and/or death receptor 5 TRAIL-R2 [69]. Ligated death receptor (DR) 4 and DR5 attach to Fas-associated death domain (FADD) then caspase-8 to become a death-inducing signaling complex (DISC) [70,71]. Lu et al. demonstrated that E-cadherin promoted apoptotic signaling via DR4/DR5 and induced caspase-8 activation at the same time increasing the sensitivity of epithelial cells to DR4/DR5-mediated apoptosis [69]. Apoptosis or programmed cell death is crucial in the regulation of malignant cell formation and treatment response [72]. It can be initiated by intrinsic or extrinsic cellular pathways. Extrinsic apoptosis pathway is initiated by death receptors which are members of tumor necrosis factors (TNFs) that are capable of regulating cell fate [73]. Several stimuli such as DNA damage, ischemia and oxidative stress lead to perturbation of mitochondrial membrane and release of cytochrome C into cytosol, where it initiates intrinsic apoptosis by inducing caspase activation [72,74]. 

## 3. Crosstalk of E-Cadherin and N-Cadherin with EMT-Related Signaling Pathways 

EMT is promoted by several transcription factors, including members of the SNAIL family, TWIST family and Zinc finger E-box-binding homeobox (ZEB) family. SNAI1, SNAI2 (SLUG), ZEB1, and ZEB2 were reported to bind directly to the E-box sequences located at the promoter region of E-cadherin gene and suppress its transcription [18,75,76,77,78]. Expression of EMT-inducing transcription factors is regulated by growth factor and cytokine signaling. Examples are the Wnt, fibroblast growth factor (FGF), Notch/delta and NF-κB pathways [58,79,80]. The PDGFR tyrosine kinase pathway has been described to regulate EMT during cancer metastasis [81,82]. N-cadherin and FGFR form a synergistic crosstalk by stabilizing FGFR-1 through the extracellular domains to support the activation of ERK 1/2 pathway and expression of MMP-9 [83,84]. Recently, re-expression of E-cadherin has been revealed to activate c-Myc and NF-κB pathway to promote the metastatic capability of tumor cells [29,85,86]. Notch signaling is a key signaling pathway that maintains the stemness of the cancer cells, induces angiogenesis, intervenes with the immune cells, and induces EMT (Figure 4) [64]. Notch-Jagged communication induces lateral induction which stabilizes the invasive phenotype and sustains the stemness of the cancer cells [87]. In contrary, Notch-Delta signaling brings different fates to two neighboring cells where one becomes a sender and the other becomes a receiver [88]. The crosstalk between E-cadherin and N-cadherin with Notch pathway and other notable EMT-related signaling pathways were summarized in Figure 4. 

### 3.1. TGF-β Pathway

Transforming growth factor β (TGF-β) pathway is well known for its role as an inducer of EMT (Figure 4). To activate cellular responses to TGF-β signals, type 1 TGF-β receptor (TβR1) needs to be phosphorylated by type 2 TGF-β receptor [89]. TβRI is activated by TGF-β signals that help in the initiation of canonical TGF-β/Smad signaling and subsequently activates Smad2 and Smad3 via C-terminal phosphorylation [90]. Phosphorylated Smad2 and Smad3 form complexes with Smad4 and translocate into the nucleus to initiate inhibition or activation of target gene transcription [90]. Smad3 activates the transcriptions of SNAI1 and SLUG by binding directly to their promoters [90]. ZEB proteins expression is modulated indirectly by TGF-β through ETS-1 [91]. 

High mobility group A2 (HMGA2) gene is activated by Smad pathway to activate SNAI1 and SLUG expression in breast cancer cell line [92]. TGF-β-activated Smad3/4 complex were found to upregulate the expression of N-cadherin through the interaction with a distinct Smad-binding component in CDH2 promoter in NSCLC cells [93]. Following the loss of E-cadherin activity, activation of TGF-β pathway is observed to suppress the nucleotide excision repair by inhibiting the transcription of XPC and DDB1 [25]. TGF-β simultaneously activates MAPK, p38, ERK1 and JNK to indirectly induce the N-cadherin expression by regulating the expression of WNT-7A via the β-catenin/T-Cell Factor (TCF) pathway [94]. Aberrant activation of TGF-β1 in hepatocellular carcinoma (HCC) leads to an increase in tumor aggressiveness by inducing EMT and stimulating the expression of Laminin-5 [95]. By using TGF-β receptor inhibitor, E-cadherin expression is elevated significantly to suppress tumor progression of HCC cells [96]. 

### 3.2. MAPK Pathway

MAPK pathways have been shown to be involved in the regulation of TGF-β/Smad signaling in EMT (Figure 4). TGF-β1 is reported to activate p38 MAPK and Jun N-terminal kinase (JNK) in lung cancer cells to increase the expression of vimentin and decrease the expression of E-cadherin [97]. N-cadherin hinders the association between JNK-associated leucine zipper protein (JLP) and p38 MAPK, thus suppressing the JNK-induced p38 MAPK pathway in the brain [98]. In breast cancer cells, p38γ MAPK is proven to induce EMT by modulating miR-200b and Suz12 via antagonizing GATA Binding Protein 3 [99]. 

Li et al. demonstrated that in breast cancer cells with high concentration of Ras pathway inhibitors, the reconstruction of E-cadherin junctions requires enforced expression of E-cadherin and inhibition of MAPK pathway. However, if Ras pathway is inhibited partially, inhibition of MAPK pathway alone was sufficient to rebuild E-cadherin junctions [100]. Interestingly, E-cadherins can activate epidermal growth factor receptor (EGFR) via interaction with tyrosine kinase receptors for EGF which subsequently induce the MAPK pathway in keratinocyte cell line [101]. MicroRNA Let-7a expression is associated with induction of EMT in prostate cancer through directly suppressing C-C chemokine receptor type 7 (CCR7) and MAPK pathway [102].

Neuropilin-1 (NRP1) is a co-receptor of vascular endothelial growth factor (VEGF) where NRP1 overexpression is reported to promote angiogenesis as well as enhance invasiveness, aggressiveness, and proliferation of cancer cells [103,104]. In oral squamous cell carcinoma, NRP1 promotes EMT by activating NF-κB pathway; conversely inhibiting NF-κB pathway reverses the NRP1-mediated EMT process [105]. Tumor-associated form of Mucin1 (tMUC1) promotes EMT by increasing the expression of VEGF through the AKT signaling pathway [106]. Zhou et al. reported that tMUC1 increases the level of NRP1 and VEGF in pancreatic ductal adenocarcinoma and this association of tMUC1 and NRP1 promotes EMT, tube formation, new vessel generation and metastasis [107]. Other than that, NRP2 has been demonstrated to upregulate the expression of ERK, MAPK, and ETS translocation variant 4 (ETV4) that promote the expression of MMP-2 and MMP-9 to suppress E-cadherin activity in oesophageal malignancy [108]. Activation of MAPK and AKT pathways induced by the overexpression of carboxyl terminus of Hsc70-interacting protein leads to an increase in the expression of SLUG and reduction in the expression of E-cadherin [109].

In colorectal cancer, pre-leukemia transcription factor 3 (PBX3) induces the MAPK/ERK pathway to improve the invasiveness of cancer cells [110]. Recently, high PBX3 expression was proved to be promoted by Wnt activation, SNAIL and ZEB1 through the indirect mediation of miR-200 in EMT induction of colorectal cancer [111]. In gastric cancer, high PBX3 expression leads to the decrease in E-cadherin expression and increase in N-cadherin, vimentin, and MMP-9 expression by promoting EMT via AKT signaling pathway [112,113].

### 3.3. JAK/STAT Pathway

Activation of the JAK/STAT pathway regulates various cell functions such as cell proliferation, differentiation, migration and apoptosis (Figure 4) [114,115,116,117,118]. N-cadherin expression in null embryonal stem cells with genetically ablated E-cadherin and human embryonic kidney cells cause an increase in STAT3 activity [119]. The homophilic interactions between Cadherin-11 and N-cadherin was responsible for the upregulation of STAT3, which subsequently leads to cell proliferation, survival, and migration [119].

Homophilic interaction of the E-cadherin resulted in an increased Rac1 and Cdc42 protein expression, which was primarily responsible for STAT3 activation (Figure 3). Inhibition of this interaction resulted in apoptosis, suggesting a role of the pathway in cell survival other than in metastasis [120]. An immunolocalization study also revealed the initial formation of phosphorylated STAT3-Y705 at E-cadherin cell junctions to be localized to the cell nucleus in head and neck squamous cell carcinoma [121]. E-cadherin has been exhibited to be involved in the induction of leukemia inhibitory factor (LIF)-STAT signaling. LIF signal is important in maintaining the stem cell characteristic of mouse embryonic stem cells (mESCs). E-cadherin is capable of forming a complex with the LIF receptor and coreceptor GP130 through its transmembrane domain, thereby stabilizing the LIF receptor complex and promoting JAK/STAT signaling [122]. The interaction of the binding domain in E-cadherin with β-catenin stimulates kruppel-like factor 4 (KLF4) and Nanog protein expression via the phosphorylation of STAT3 [123]. Interestingly, it was revealed that N-cadherin, in the absence of E-cadherin, can compensate for E-cadherin-dependent STAT3 phosphorylation in mESCs [123].

Knockdown of STAT3 in colorectal cancer (CRC) cells showed a significant increase in E-cadherin expression and decreased N-cadherin and vimentin expression. Evidence indicated that STAT3-induced CRC cell invasion and E-cadherin downregulation were dependent on ZEB1 and could be rescued by downregulation of ZEB1 [124]. In another study, IL-27 resulted in activation of STAT1 and STAT3 in a JAK-dependent manner as well as decreased expression of E-cadherin by upregulating SNAIl in human NSCLC [125]. Metformin, an anti-diabetic drug, successfully suppressed IL-6 induced STAT3 activation and EMT in lung cancer cells. [126].

### 3.4. Hedgehog Pathway

Aberrant activation of HH signaling mediates EMT in various cancers by altering the expression of E- and N-cadherin, as well as other mesenchymal markers (Figure 4). Li et al. discovered that a dominant negative of SNAI1 in GLI1 transformed epithelial cells increases E-cadherin levels and weakens their ability to grow independent of attachment [127]. Xu et al. also stated that non-canonical GLI1 activation of the HH pathway and MAPK pathway could lead to the decrease of E-cadherin expression in thyroid cancer cell lines, which subsequently lead to BRAF mutation-induced SNAI1 expression [128]. Besides that, sonic hedgehog (SHH)/GLI1 expression induced a significant upregulation of expression level of S100A4, a member of the S100 gene family and a key EMT molecular marker in pancreatic cancer, while E-cadherin was markedly reduced [129]. S100A4 gene had been shown to promote the expression of TWIST and SNAIl along with other mesenchymal markers during EMT [130]. As there is no evidence indicating direct activation of SNAI1 by GLI1, these data suggest a possible connection between SHH and S100A4 during EMT in pancreatic cancer cells [129].

Casticin or SMO-inhibitor (cyclopamine) treated ovarian cancer cell line showed notably increase in E-cadherin expression and reduction in N-cadherin expression [131]. Resveratrol and cyclopamine treatment upregulated E-cadherin and downregulated GLI1, SNAI1 and N-cadherin in gastric cancer cell line [132]. Similarly, a SMO knockdown human pancreatic cancer stem cell demonstrated a decrease in the expression of E-cadherin and increase in the expression of SNAI1 and N-cadherin [133]. In esophageal adenocarcinoma (EAC), GLI-inhibitor (Gli-i) upregulated E-cadherin expression while downregulating N-cadherin and β-catenin expression by inhibiting GLI1 and GLI2 transcriptional activity. In addition, a combination of AKT-inhibitor (AKT-i) and N-SHH rescued the effect induced by AKT-i, which was observed when using Gli-i instead of AKT-i. The study demonstrated that SHH/GLI1 signaling in EAC may potentially regulate EMT via AKT pathway [134].

HH signaling has shown differing effects on E-cadherin as one study has highlighted that inhibition of HH signaling with cyclopamine caused the loss of E-cadherin and relocalization of ZO-1 [135]. Another study has revealed that GLI1 promoted the redistribution of E-cadherin toward the cell membrane [136]. Liao et al. found that abnormal GLI1 activation increased the expression of SNAI1 and E-cadherin in ovarian cancer cells, indicating that E-cadherin may be regulated via a different molecular network other than SNAI1 [137]. 

### 3.5. Wnt Pathway

In general, cadherins are considered as negative regulators of the Wnt pathway by sequestering β-catenin from T-cell Factor (TCF) family transcription factors to the plasma membrane (Figure 2, Figure 3 and Figure 4). When Wnt proteins bind to the Frizzled family on the cell surface, Wnt signaling is initiated. The signal is then transduced to β-catenin via various cytoplasmic relay components to stabilize cytoplasmic β-catenin. β-catenin then accumulates and enters the nucleus to activate transcription of Wnt target genes by interacting with LEF/TCF proteins [138]. As β-catenin/TCF encodes c-MYC protein, the disturbed sequestering of β-catenin and aberrant activation of Wnt pathway can lead to tumorigenesis [139,140]. In the absence of Wnt signals, the soluble cytoplasmic pool of β-catenin is recruited into the Axin/APC degradation complex, while the adhesion function is based on a cadherin-bound, stable pool at the membrane [141].

E-cadherin expression promotes extranuclear β-catenin translocation hence suppressing the Wnt pathway (Figure 2) [29]. It is evident that crosstalk between cadherins and the canonical Wnt/β-catenin pathway exist and are linked in multiple ways. Wnt signaling had been proven to downregulate E-cadherin in embryonic mouse brain [142]. It was shown that Wnt signaling promoted EMT, increased N-cadherin, TWIST, SLUG and decreased E-cadherin expression. The decrease in E-cadherin can be explained as canonical Wnt signaling prevents glycogen synthase kinase 3β (GSK3β) from phosphorylating SNAI1, allowing it to accumulate and repress E-cadherin expression. As decreased E-cadherin expression leads to higher availability of β-catenin in the cytoplasmic pool, this would ultimately lead to an increase in LEF/TCF transcriptional activity and Wnt signaling [143]. 

Several mechanisms had been reported on how cadherins regulate Wnt signaling. One mechanism involves the formation of a phospho-destruction complex at the site of adherens junction. E-cadherin can inhibit β-catenin signaling by inducing its N-terminal phosphorylation. Subsequently, this phosphorylated β-catenin will localize to cadherin-containing cell contacts through its association with the axin-phosphodestruction complex, thereby limiting Wnt signaling [144]. Another mechanism for cadherins to regulate Wnt signaling is via utilizing proteases such as caspase 3, presenilin, ADAM10 or calpain to cleave E- and N-cadherin so that β-catenin will be released to activate Wnt/β-catenin target genes [143]. An alternative mechanism may require the involvement of tyrosine kinases and phosphatases that either positively or negatively regulates the cadherin-catenin complex depending on the site of phosphorylation [145]. 

Protein kinases CK2 and GSK3β could phosphorylate Ser684, Ser686 and Ser692 in the cadherin cytoplasmic domain, which strengthens the E-cadherin/β-catenin complex and cell–cell adhesion, thereby limiting Wnt signaling [146]. On the contrary, phosphorylation of a critical amino acid Tyr-860 of N-cadherin by Src tyrosine kinase induces the dissociation and nuclear localization of β-catenin which involves the direct association between the cytoplasmic tail of cadherin and β-catenin [147]. In transitional cell bladder carcinoma (TCC) cell lines expressing E-cadherin, activated β-catenin was insufficient to upregulate the transcription of the β-catenin/TCF-dependent promoter. Conversely, TCC cell lines lacking E-cadherin revealed marked induction. Similar findings were obtained upon re-expression of E-cadherin in the corresponding TCC cell lines, whereby promoter activation by β-catenin was inhibited followed by the co-transfection of E-cadherin [148]. 

Interestingly, GLI1 of the HH signaling pathway was identified as a crucial regulator of the β-catenin switch in transformed epithelial cells. GLI1 was found to promote the localization of β-catenin in cells by suppressing E-cadherin through SNAI1 expression. Transformation suppressor activity of E-cadherin only function upon binding to β-catenin, supported by the evidences that transfection of E-cadherin containing a deleted amino acid sequence in the C terminus failed to react with *β*-catenin but maintained cell–cell adhesion and hence was unable to activate transformation by GLI1 [149].

Conversely, it was demonstrated that cadherin may serve as a positive regulator of Wnt/β-catenin during EMT by providing signaling-competent β-catenin after endocytosis, despite various studies indicating cadherin as a negative regulator of Wnt/*β*-catenin signaling [150]. This is proven when dynasore, an inhibitor of dynamin-dependent endocytosis, reduced Wnt pathway readout despite transfection of cells with β-catenin [150]. It is possible that the β-catenin derived from the cadherin complex may be different compared to the newly translated β-catenin that has never attached to cadherin and could not regulate transcriptional activity [150]. Wnt signaling may form a positive feedback loop which augments its own transcriptional activity through cadherin [150].

Lastly, the final mechanism involves the direct interaction of the N-cadherin with low-density lipoprotein receptor (LDLr)-related protein 5 (LRP5), a co-receptor for Wnt ligands [151]. In osteoblast, the interaction of N-cadherin with LRP5 recruits axin, forming an axin-LRP5 complex that leads to the increased ubiquitination of β-catenin and the reduction in osteoblast differentiation [151]. A further study showed that a competitive peptide, which effectively disrupts LRP5/6-N-cadherin interaction, was able to activate Wnt signaling and bone formation [142]. 

### 3.6. Hippo-YAP/TAZ Pathway

The Hippo-Yes-associated protein (YAP)/transcriptional co-activator with PDZ-binding motif (TAZ) pathway is an evolutionary conserved signaling pathway that plays a crucial role in regulating organ size and tumorigenesis by moderating the balance between cellular proliferation and apoptosis [152]. In the classical Hippo-YAP pathway of mammals, the MST1/2 kinases phosphorylate the hydrophobic motif and activates a second set of Large Tumor Suppressor Kinase (LATS) 1/2 [152,153]. The activated LATS1/2 then phosphorylates the growth-promoting transcriptional co-activator YAP1/TAZ, leading to the sequestration and retention of these proteins in the cytoplasm [154]. Inhibition of the Hippo-YAP signaling pathway promotes the translocation of YAP1/TAZ into the nucleus, thereby allowing the activation of the downstream genes (Figure 4) [154].

Generally, α-catenin does not directly interact with cadherins but links the adherens junction or cadherin-catenin complex to the actin cytoskeleton by binding to β-catenin [154]. E-cadherin is crucial in regulating YAP nuclear localization in a cell density-dependent manner, and the mechanism is dependent on α- and β-catenin interaction with E-cadherin [155]. This showed that the cadherin-catenin complex, but not E-cadherin itself, was responsible for the regulation of Hippo-YAP1 signaling [155].

Moesin-ezrin-radixin-like protein (Merlin) can physically interact and suppress YAP/TAZ activity through its nuclear export sequences [156]. Depletion of Merlin relieves the suppression of YAP/TAZ nuclear localization of ROCK3-constitutively active mutant cells without affecting the circumferential actin belt contractility [156]. During high tension circumferential actin belt state, the interaction between Merlin and E-cadherin is abrogated, which releases Merlin to bind and export YAP/TAZ from the nucleus with its own nuclear export sequence [156]. Similarly, depletion of Merlin and its adaptor protein Na^+^/H^+^ exchanger regulatory factor (NHERF) relieved the E-cadherin ligation-mediated inhibition [155]. It was reported that YAP1 co-activated transcription with β-catenin and inhibition of β-catenin led to the inactivation of YAP1 and reduced cellular proliferation in human hepatoblastoma samples [157]. Similarly, α-catenin was able to limit the YAP1 transcriptional activity in human squamous cell carcinoma [158] and control the proliferation of both cardiomyocyte [159] and epidermal stem cell [152], further proving the role of cadherin-catenin complex in the regulation of Hippo-YAP signaling pathway.

Hippo-YAP pathway was modulated by cell shape and mechanotransduction [160,161]. The ability of YAP/TAZ to sense mechanical cues such as cytoskeleton tension was proven when TAZ degradation was observed in mammary epithelial cells grown on soft matrices or treated with Rho, F-actin and actomyosin tension inhibitors [162]. It was revealed that high mechanical stress promotes YAP/TAZ-mediated proliferative activity, whereas low mechanical stress inhibited the activity of YAP/TAZ by F-actin-capping and -severing proteins independent of the Hippo-YAP signaling cascade [163]. However, other studies have found that perturbations in actin such as cytoskeleton reorganization, extra F-actin polymerization and altered expression of F-actin would regulate YAP/TAZ activity through the Hippo pathway by LATS [164]. It is possible that different parts of the actin cytoskeleton such as focal adhesion–associated stress fiber or cell junction may regulate the Hippo-YAP signaling differently [161].

Hippo-YAP signaling may affects the expression of E- and N-cadherin by regulating the EMT process. In gastric cancer cells, SOX9 promotes EMT by activating the Hippo-YAP pathway and SOX9 silencing rescued the expression of E-cadherin and downregulated mesenchymal markers N-cadherin, vimentin and SNAIl [165]. WW and C2 domain containing protein-3 (WWC) was demonstrated to hinder EMT in lung cancer cells by initiating the Hippo pathway [166]. In cholangiocarcinoma, overexpression of YAP enhances tumorigenesis and metastasis both in vivo and in vitro. It was found that YAP was able to form a regulatory network with miR-29c, Insulin-like growth factor 1 (IGF1), AKT and Gankyrin to induce EMT and promote carcinogenesis by activating AKT pathway [167]. Similarly, ectopic expression of TAZ promotes cell proliferation, reduces cell contact inhibition and enhances EMT, where phosphorylation of TAZ by LATS reverses these effects [168]. 

## 4. Therapeutic Implication Targeting EMT

### 4.1. shRNA and miRNA

Introduction of short hairpin RNA (shRNA) and truncated E-cadherin (without the ectodomain) have been used to suppress E-cadherin function and induce the expression of TWIST and ZEB1 in human breast epithelial cells [24]. The miR-145 was reported to affect the translation of N-cadherin by directly targeting its 3′-UTR [169]. In a study, miR-145 mimics-transfected human lung adenocarcinoma cells showed remarkable reduction of N-cadherin expression at both transcriptional and translational levels [169]. However, the transcriptional and translational expressions of N-cadherin were not significantly affected by miR-145 transfection in prostate cancer cells [49]. 

### 4.2. Small Molecules and Tyrosine Kinase Inhibitors

Treatment with FGFR kinase inhibitor PD173074 reduced migration and invasion of PyVmT-N-cadherin cells [51]. Metformin exhibits TWIST dependent anti-tumor properties by inhibiting N-cadherin and NF-κB signaling without affecting E-cadherin in wild-type N-cadherin expressing tumor cells including prostate, bladder, and kidney [170]. On the other hand, pirfenidone, an orally active small molecule compound is capable of inhibiting EMT and fibrosis by suppressing the MAPK pathway with upregulated E-cadherin expression in kidney cells [171].

### 4.3. Monoclonal Antibodies

E- and N-cadherin contain a conserved HAV sequence, where linear peptides containing this motif are able to suppress cadherin-dependent processes [172]. Besides, antibodies targeting the HAV sequence can disorganize cadherin-dependent cell adhesion [172,173]. Tanaka et al. showed that monoclonal antibodies against the ectodomain of N-cadherin could suppress cell proliferation, cell adhesion and inhibit the invasiveness of prostate tumor cells *in vitro* [174]. The same antibodies also suppressed the cell proliferation and metastasis *in vivo,* decreased the activity of AKT kinase, and reduced the secretion of IL-8 [174]. 

### 4.4. Natural Compounds

#### 4.4.1. Curcumin

Curcumin is the major bioactive compound in the rhizome of *Curcuma longa* L. also known by its conventional name as turmeric, which belongs to the family Zingiberaceae. Curcumin has been shown to exert anti-cancer activity in addition to its role as an anti-oxidant, anti-infective, wound healing, hepatoprotective and neuroprotective activity [175,176,177,178]. This compound can modulate multiple intracellular molecular targets in several preclinical disease models, including cancer and cancer stem cells [175,176,177,178,179,180].

Curcumin was reported to inhibit breast cancer stem cell migration by decreasing nuclear translocation of β-catenin and increasing E-cadherin/β-catenin complex formation in the cytosol thereby suppressing EMT [181]. Curcumin suppressed HeLa and SiHa cervical carcinoma cells by inhibiting the TGFβ pathway and downregulating the expression of cyclinD1, p21 and Pin1, TGF-βRII, p-Smad-3, Smad-4, SNAI1, and SLUG [182]. Besides, curcumin significantly inhibited TGF-β stimulated Panc1 pancreatic cancer cells proliferation, invasion and migration, induced apoptosis and reversed EMT by modulating the SHH-GLI1 signaling pathway [183]. In triple negative breast cancer (TNBC) cells, curcumin reversed doxorubicin induced EMT by the downregulation of the TGFβ and phosphoinositide-3-kinase (PI3K)/AKT signaling pathway [184]. 

Bisdemethoxycurcumin (BDMC) is another bioactive compound of curcumin that has been shown to inhibit invasion, metastasis and tumor growth in multiple cancers. BDMC suppressed highly metastatic NSCLC cells proliferation and TGFβ induced EMT by downregulating Wnt inhibitory factor 1 (WIF-1) [185]. In another study, curcumin was employed to inhibit TGF-β-induced EMT by downregulating Smad2/3 signaling pathway in BCPAP thyroid cancer cells [186]. In addition, curcumin was found to abrogate cancer associated fibroblast-induced prostate cancer cells invasion by downregulating monoamine oxidase A (MAOA)/mammalian target of rapamycin (mTOR)/hypoxia-inducible factor-1α (HIF-1α) signaling pathway [187]. Curcumin suppressed EMT and angiogenesis by inhibiting c-met/PI3K/AKT/mTOR signaling pathway metastasis and induced apoptosis in lung cancer cells in vitro and in vivo [188]. In a nude mice xenograft lung tumor model, curcumin significantly inhibited HGF-induced tumor growth and EMT [188]. Curcumin loaded selenium nanoparticles was found to significantly downregulate EMT-metastasis-associated proteins and promote apoptosis of HCT116 CRC. These nanoparticles also remarkably decreased tumor burden and increased survival of Ehrlich’s ascites carcinoma (EAC)-bearing mice [189,190]. In glioma LN229 and U251 cells, curcumin reversed the EMT process induced by γ-irradiation via the suppression of GLI1 and the upregulation of Suppressor of Fused Homolog (SUFU), as well as by suppressing the HH signaling pathway both in vitro and in vivo [191]. To nude mice carrying intracranial glioma tumor, curcumin was injected and induced MET while suppressing tumor growth [191]. The Enhancer of Zeste Homolog-2 (EZH2) subunit of Polycomb Repressive Complex 2 (PRC2) was recently identified as a key player regulating drug resistance [192]. EZH2 mediates interaction with several long non-coding RNAs (lncRNAs) to modulate EMT and cancer stemness, a phenomena commonly associated with drug resistance [193]. 

In gemcitabine-resistant pancreatic ductal adenocarcinoma cells (BxPC3-GemR cells), curcumin sensitized the cells by modulating the PRC2-PVT1-cMyc axis in vitro and inhibited the growth of BxPC3-GemR cells in a xenograft mouse model [194]. Gemcitabine alone, curcumin alone or combinations of gemcitabine and curcumin significantly reduced tumor growth [194]. Recently a synthetic curcumin analog, PAC (u4-hydroxy-3-methoxybenzylideneN-methyl-4-piperidone) exhibited higher bioavailability and potent anti-cancer activity and was demonstrated to downregulate estrogen receptor (ER) and EMT in breast cancer cells in vitro and in vivo [195,196]. PAC administration inhibited the growth of subcutaneously implanted MDA-MB-231 breast cancer cells in a nude mice model and was associated with downregulation of AKT and ERK1/2, up-regulated E-cadherin, while it down-regulated N-cadherin, vimentin, and TWIST1 [195,196]. Similarly, in CRC, PAC was shown to suppress EMT and was associated with concomitant suppression of MEK/ERK, JAK2/STAT3, and AKT/mTOR signaling pathways both in vitro and in vivo [197]. PAC inhibited colorectal tumor growth in a nude mice model [197].

Fibroblast activation protein α (FAPα) vaccine in combination with curcumin was shown to significantly inhibit TNFα-induced EMT in melanoma cells by targeting indolamine-2,3-dioxygenase, inhibit tumor growth and prolong the survival of mice implanted with melanoma cells [198]. Curcumin was found to upregulate the expression of miR-101, miR-141, miR-200b, miR-200c, and miR-429 in 5-fluorouracil (5-FU) resistant cell lines. In contrast, 5-FU treatment did not affect the EMT suppressive miRs in 5-FU resistant cells [199]. Interestingly EMT suppressive miR-34a was upregulated in HCT-116-5-FU cells and not in SW480-5-FU cells. In a murine xenograft mouse model, curcumin either alone or in in combination with 5-FU upregulated tumor suppressive miR-200c and suppressed tumor growth compared to control group [199]. Systemically administered curcumin markedly decreased the expression genes associated with EMT and was shown to inhibit oral carcinogenesis induced by 4-nitroquinolone-1-oxide (4-NQO) in albino rats [200].

#### 4.4.2. Resveratrol

Several studies have implicated the resorcinol derivative resveratrol, (3,5,4′-trihydroxy-trans-stilbene) in the regulation of EMT. In pancreatic carcinoma cells, resveratrol was shown to inhibit proliferation, invasion, migration and EMT by suppressing PI3K/AKT/NF-κB pathway thereby modulating the expression of EMT-related genes [201,202]. Resveratrol was found to suppress the expression of EMT-inducing transcription factors such as SNAI1 and SLUG with concomitant increase in expression of E-cadherin along with suppression of vimentin and fibronectin in lung cancer cells [203]. Resveratrol inhibited stemness, metabolic reprogramming in cancer stem cells and EMT in nasopharyngeal carcinoma cells by increasing the expression of miR-145 and miR-200c, and was associated with reactivation of tumor suppressor p53 thereby suppressing EMT [204]. In a recent study, resveratrol was found to modulate miR-200c expression and prevent proliferation, invasion and EMT in HCT-116 CRC [205]. Resveratrol blocked colony formation, invasiveness and metastatic capacity of head and neck cancer cells by modifying EMT markers SLUG, ZEB1, E- and N-cadherin, and vimentin. Furthermore, resveratrol was also shown to modulate stemness associated genes Oct-4, nanog, nestin in tumor-initiating cells [206]. Oral gavage of resveratrol to mice repressed tumor growth, as well as tumor stemness and EMT markers expression in a xenograft mice tumor model [206]. 

Resveratrol abrogated EGF- and TGF-β-induced EMT in breast cancer cells [207,208]. Adriamycin was shown to induce E-cadherin-mediated cell–cell adhesion by increasing the expression of E-cadherin and β-catenin while decreasing the expression of Mucin 1 in YMB-S breast cancer cells [209]. In adriamycin-resistant MCF7/ADR breast cancer cells resveratrol sensitized the cells to doxorubicin and promoted cell apoptosis [210]. Resveratrol reversed EMT in MCF7/ADR cells by inhibiting the connection between SIRT1 and β-catenin [210]. In another study, resveratrol in combination with salinomycin (Wnt inhibitor) significantly suppressed canonical Wnt signaling molecules and markers of EMT in ER positive breast cancer cells [211]. 

Furthermore, resveratrol induced β-TrCP-mediated TWIST1 degradation to attenuate AKT inhibitor MK-2206-induced EMT in breast cancer and that AKT1/PKBα serves as a negative regulator of EMT and breast cancer metastasis [212]. In gastric cancer SGC-7901 cells resveratrol inhibited EMT by suppressing the HH pathway [132]. Yang et al. reported that resveratrol inhibited metastasis-associated lung adenocarcinoma transcript 1 (MALAT1) mediated EMT, invasion, and migration, providing new indication for understanding the anticancer mechanism of resveratrol in BGC823 gastric cancer cells [213]. In oral squamous cell carcinoma CAL27 cells, resveratrol inhibited EMT, invasion and migration by suppressing EMT-inducing transcription factors and induced mitochondrial pathway mediated apoptosis [214]. Remarkably, a novel diaryl-substituted imidazole analog of resveratrol was shown to significantly inhibit EMT makers expression in ovarian cancer cells [215]. In doxorubicin resistant gastric cancer (SGC7901/DOX) cells, resveratrol treatment improved doxorubicin sensitivity, mitigated the aggressive biological features, as well as promoted cell apoptosis in vitro and in vivo by inhibiting EMT and PTEN/AKT pathway [216]. 

TWIST1 and SNAI1 can be downregulated upon resveratrol treatment and was associated with concomitant suppression of the Wnt/β-catenin signaling pathway in glioma stem cells [217]. Furthermore, resveratrol analogs such as DHS (4,4′-dihydroxy-trans-stilbene), HPIMBD (4-(E)-[(4-hydroxyphenylimino)-methylbenzene,1,2-diol]), pterostilbene which is a natural potent analog of resveratrol and other metabolites of resveratrol such as RSV-3-O-sulfate, RSV-3-O-glucuronide, and RSV-4′-O-glucuronide were shown to possess anti-metastatic and EMT inhibitory activity in a variety of cancer cells [218,219,220]. Resveratrol sensitized 5-FU resistant colorectal cancer cells, induced apoptosis and inhibited EMT phenotype by upregulating intercellular junctions and E-cadherin expression with concomitant downregulation of the NF-κB pathway [221]. Resveratrol inhibited TGFβ-induced EMT in colorectal cancer cells by downregulating TGFβ/Smads signaling pathway and reduced the rate of lung and hepatic metastases in mice orthotopic mouse model [222]. In osteosarcoma Saos-2 cells, resveratrol downregulated HIF-1α protein expression and hypoxia induced EMT [223]. Treatment with resveratrol was shown to reverse EMT induced by lipopolysaccharide by modulating in vitro cell motility and invasiveness through the HH pathway in prostate cancer cells [224]. In another study, resveratrol prevented cisplatin-induced EMT in both parental and cisplatin resistant ovarian cancer cells in an apoptotic-independent manner [225].

#### 4.4.3. Honokiol

Honokiol (3,5-di-(2-propenyl)-1,1-biphenyl-2,2-diol) is a phenylpropanoid molecule, a biaryl-type lignan present in the genus magnolia [226,227]. A recent report shows that honokiol has potent anti-cancer and anti-EMT activity in metastatic melanoma cells by modulating the β-catenin/MITF axis. In a melanoma peritoneal metastasis mouse model, honokiol was shown to inhibit highly metastatic dissemination of melanoma in mice [228]. Honokiol loaded pH-sensitive polymer-doxorubicin conjugate micelles (HNK) was shown to inhibit the growth of TNBC cells. In a pulmonary metastasis mouse model, HNK in combination with doxorubicin synergistically inhibited the tumor growth and metastasis [229]. Interestingly, honokiol inhibited steroid receptor coactivator-3 (SRC-3) with concomitant downregulation of EMT markers in bladder cancer cells [230]. In NSCLC A549 and H460 cells, honokiol inhibited TNFα and TGFβ induced EMT by targeting and downregulating c-FLIP, N-cadherin, SNAIl proteins, and pSmad2/3 and NF-κB [231].

Moreover, honokiol was shown to inhibit leptin induced EMT and mammosphere formation with concomitant decrease in expression of stemness factors, OCT4, Nanog and STAT3/ZEB1/E-cadherin axis in breast cancer cells [232,233]. Honokiol was reported to downregulate TGFβ and *N*-methyl-*N’*-nitro-*N*-nitrosoguanidine (MNNG) induced EMT and Calreticulin (CRT). Furthermore, honokiol significantly suppressed MNNG induced gastrointestinal tumor growth, metastasis and over-expression of CRT in mouse model [234]. Similarly, in MNNG induced tumor model, honokiol was shown to suppress tumor progression locus 2 (Tpl2) with concomitant suppression of EMT markers [235]. In renal cell carcinoma, honokiol suppressed metastasis via dual-blocking EMT and cancer stemness by inhibiting ZEB2 and upregulating miR-141 [236]. Interestingly, honokiol was shown to be able to cross the blood brain barrier and blood-cerebrospinal fluid barrier [237]. Honokiol suppressed glioblastoma EMT markers and dose-dependently inhibited TNFα induced VCAM1 expression in brain microvascular endothelial cells [238]. 

#### 4.4.4. Other Natural Compounds

(-)-Epigallocatechin-3-gallate (EGCG) is the main polyphenol in green tea and was proven to demonstrate anti-oxidative, anti-inflammatory, anti-metastatic, and anti-carcinogenic functions through different pathways [239,240,241]. Wu et al. demonstrated that EGCG inhibited the adhesion ability and cells invasion in a concentration-dependent manner while increasing the expression of E-cadherin in melanoma cells [242]. Administration of EGCG into intestinal cancer mouse model led to elevation of E-cadherin protein levels and decrease of protein levels of c-Myc, nuclear β-catenin, protein kinase B, and ERK1/2, indicating that EGCG inhibits intestinal tumorigenesis through suppression of the AKT and ERK signaling cascade [243]. Downregulation of N-cadherin and inactivation of AKT signaling were also observed in bladder carcinoma cells that received a continuous intake of EGCG [244]. EGCG administration also suppressed the levels of TGF-β1, TGF-βRI and phosphorylation-Smad3 and relieved prostatic EMT by increasing the expression of E-cadherin in rat models [245]. A study also revealed that EGCG administration inhibits inflammation and increases the E-cadherin protein level in asthma via elevating the expression of Phosphatase and Tensin homolog (PTEN) to attenuate PI3K/AKT signaling activation [246]. 

S-allylcysteine (SAC) and S-allylmercaptocysteine (SAMC) are water soluble anticancer compounds isolated from garlic [247,248]. SAC and SAMC inhibited the proliferation and invasiveness of prostate cancer cells in vitro and in vivo with low toxicity while inducing MET and restored expression of E-cadherin [247,248]. An oxindole compound extracted from *Periplaneta americana*, ZL170 was demonstrated to inhibit EMT in TNBC by suppressing TGFβ/BMP pathway [249]. Neferine is a major benzylisoquinoline alkaloid isolated from the seed embryo of lotus [250]. Neferine possesses a variety of therapeutic effects including anti-cancer, anti-diabetic, anti-aging, anti-inflammatory and enhances the chemosensitiviy as previously reviewed by Asokan’s group [251]. Neferine treatment increases oxaliplatin chemosensitivity in HCC by suppressing EMT via elevating E-cadherin expression and decreasing Vimentin, SNAIL and N-cadherin expression [252]. 

Luteolin ((3,4,5,7-tetrahydroxy flavone) is a natural flavonoid that is extensively found in many fruits and vegetables, for example chrysanthemum flowers, sweet bell peppers, carrots, broccoli, and onion leaves [253,254,255]. Luteolin has been reported to reverse EMT by involving the upregulation of E-cadherin expression and downregulation of N-cadherin, SNAIL, and vimentin expression [255]. In prostate cancer, luteolin increases the expression of E-cadherin by inhibiting mouse double minute 2 (MDM2) through AKT pathway [256]. A recent study also reported that luteolin impedes TGF-β1-induced EMT via elevating the expression of miR-203 and suppressing Ras/Raf/MEK/ERK signaling in breast cancer [257,258]. Luteolin has been reported to inhibit or reverse EMT in several cancers through regulating a variety of signaling pathways, such as Notch signaling in gastric cancer [259], HIF-1α/VEGF and β3 integrin/FAK signaling pathways in melanoma [260], and PI3K/AKT-NF-κB-Snail pathway in lung cancer [261], making luteolin an interesting therapeutic candidates as complementary medicine for a wide range of cancers [262].

Kaempferol (3,5,7-trihydroxy-2-[4-hydroxyphenyl]-4H-1-benzopyran-4-one) is a potent flavonoid that possesses anti-cancer, anti-inflammatory, anti-aging, antiallergic and antiplatelet aggregation properties [263]. Kaempferol suppresses tumor angiogenesis and VEGF expression via modulating HIF dependent and independent pathways [264]. Jo et al. demonstrated that kaempferol reverses TGF-β1-induced SNAIl induction and restores E-cadherin expression through selectively impeding AKT1-mediated phosphorylation of Smad3 at Thr179 residue in human NSCLC cells which weakens Smad3 binding to the SNAIL promoter [265]. In concordance to their findings, Liang et al. found that kaempferol reduced the expression of mesenchymal markers and inhibited EMT by the TGF-β-dependent signaling in NSCLC [266].

## 5. Challenges to Translating the Preclinical Research from Bench to Bedside

Although some of the compounds mentioned above had demonstrated remarkable anti-proliferative activities in in vitro and in vivo models, and the mode of action had been linked to the suppression of EMT, the challenges of moving these along the drug development pathway of undergoing pre-clinical and clinical trials are palpable. Thus far, there have been some natural compounds derived anticancer agents that attained the ultimate status of being registered and being an FDA-approved chemotherapeutic drug, including the success stories of paclitaxel and vincristine. In the case of paclitaxel (Taxol^®^), it is a member of the taxane class of compounds, derived from the bark of the Pacific Yew Tree [267]. As an anticancer agent which works by stabilizing microtubules and targeting tubulins, paclitaxel arrests cell cycle at the G2/M phase and inhibits the function of the apoptosis inhibitor protein B-cell Leukemia 2 (Bcl-2) [268]. Due to its strong hydrophobicity, paclitaxel has been formulated in different forms including in castor oil, or bound with albumin, although newer drug formulations using nanoparticles have shown to be decrease the toxicity and enhance its delivery and distribution (reviewed comprehensively by [269]); making it effective for the current clinical use for various forms of cancers. On the other hand, vincristine is a vinca alkaloid derived originally from *Catharanthus roseus* (the Madagascan periwinkle). It is used mainly for chemotherapy against leukemias, lymphomas, neuroblastoma, and sarcomas. Vincristine acts via binding to tubulin and blocking metaphase in actively dividing cells [270]. As of 2013, a total of 100 natural products in their native forms as well as their derivatives were in the process of clinical trials [271].

One of the reasons for the failure of many compounds to make it to the next stage of the drug development pipeline is the issue related to the bioavailability of these compounds [272]. Specifically, some of the problems causing the lack of bioavailability are low absorption, rapid metabolism, as well as rapid systemic elimination [273]. A case in point is curcumin, whereby despite its proven bioactive potential, its clinical usage is hampered by its relative instability and low solubility, in addition to being rapidly metabolized and thus having poor bioavailability [274]. Many dietary flavonoids and other polyphenols have poor oral bioavailability where it is believed that extensive conjugation of the free hydroxyl groups is the main reason for the low oral bioavailability [275]. A possible strategy to overcome the poor bioavailability would be to modify the chemical structures of the promising compounds in order to enhance their absorption, distribution, and excretion characteristics [276]. Besides that, other approaches to enhance bioavailability include the use of adjuvants, liposomal encapsulation, and phospholipid complexes [276]. A classic study of concomitant administration of a 2g dose of curcumin with 20 mg piperine improved the pharmacokinetics and bioavailability of curcumin in human subjects by 2000% [277]. Another inherent problem of many natural compounds is their undesirable toxicity effects to non-targeted or non-cancerous cells and tissues. In some cases, this may be circumvented by structural alterations to the molecules, or by alternative formulations. Conjugation to other small molecules may sometimes provide the necessary modification that reduce the toxic side-effects if these plant-derived compounds. An instance is the synthesis of curcumin analogs which successfully prevented lung carcinogenesis via activating the Nrf2 pathway, thus achieving the goal of chemoprevention [278]. A review of the important synthetic derivatives and analogs of curcumin and their potential pharmacological activities was provided by Bukhari and co-authors [279].

Although resveratrol possesses multiple bioactivities, this compound shows unique pharmacokinetic and pharmacodynamics characteristics that pose limitations for its applications. Resveratrol is extensively metabolized in the intestine (Phase I metabolism) and liver (Phase II metabolism) and is rapidly eliminated and therefore shows poor bioavailability [280]. In the intestine, this compound undergoes a first-pass glucuronidation and sulfate conjugation of the phenolic groups and hydrogenation of the aliphatic double bond, whereas in the liver, Phase II metabolism of resveratrol and its metabolites occurs. It has been shown that most of the plasma resveratrol metabolites have very low bioactivity. The rate-limiting step in resveratrol bioavailability has been suggested to be due to the remarkably rapid sulfate conjugation by the intestine/liver [281]. To this end, many methodologies have been employed with hope to improve the bioavailability of resveratrol. These include the use of solid dispersions, micelles, lipid nanocarriers or liposomes, nanoemulsions, polymeric nanoparticles and nanocrystals as carrier systems, all of which are summarized comprehensively in a recent review [282]. 

Advances in nanotechnology have allowed more efficient and targeted drug delivery. There are many instances whereby nanoparticle carriers have allowed the pharmacotherapeutic potential of these compounds to be enhanced. A vast option of nanotechnological solutions have been investigated for their suitability as nanocarriers for bioactive natural compounds including polymeric nanoparticles, solid lipid nanoparticles, nanogels, drug-functionalized nanodiamonds, nanosuspensions and nanoemulsions [283]. For instance, the pharmacokinetics profile of curcumin-loaded polylactic-co-glycolic acid (PLGA) and PLGA-polyethylene glycol (PEG) (PLGA-PEG) nanoparticles was evaluated, and was demonstrated to have slow release in vitro, and to have increased the curcumin mean half-life and bioavailability up to 15.6- and 55.4-fold respectively in rats [284]. In a recent study which compared the oral bioavailability of the newer, US FDA-approved Eudragit^®^ RLPO (ERL) nanoparticles with that of PLGA and PCL (polycaprolactone) nanoparticles, the ERL nanoparticles were shown to exhibit the most rapid total release of curcumin; thus, the authors concluded that ERL was the most promising carrier for the oral delivery of curcumin [285]. On another note, a recent study by Thipe and coworkers showed convincing evidence that conjugating resveratrol with gold nanoparticles not only enhanced the bioavailability of resveratrol but also provided synergistic anti-tumor effects due to the natural anti-angiogenic effects of gold nanoparticles or AuNPs [286]. Separately, an earlier study revealed that resveratrol-capped gold nanoparticles (Rev-AuNPs) were capable of impeding invasive properties in human breast cancer cells via targeting the MMP-9 and COX-2 expression [287]. 

To date, there has been a steady increase in the number of research studies that employ some of the various abovementioned strategies in overcoming the challenges of drug delivery for these natural compounds. We remain hopeful that some of these star compounds which target the EMT pathway would emerge eventually as effective chemotherapeutic drugs approved for clinical use.

## 6. Conclusions

Multiple natural compounds with anticancer activity has been shown to be able to inhibit EMT by suppressing some of the key molecules or pathways involved in EMT, such as N-cadherin, and the Hedgehog signaling pathway, and/or increasing the E-cadherin/β-catenin complex. As discussed in this review, plant-derived natural molecules including curcumin, resveratrol, and honokiol, as well as synthetic molecules such as shRNA and engineered antibodies possessed potent activity that can suppress EMT. Functional and mechanistic studies using in vitro studies provided the all-important evidences for their potential usage and it will be interesting to probe how this will translate into preclinical animal studies that will eventually justify their usage in human clinical trials. Nonetheless, more studies are warranted to test the bioavailability of the formulations as well as the safety profile in terms of systemic toxicity in humans. Taken together, these compounds have promising potential, especially in targeting and attenuating the transformation of cancer cells into more aggressive, invasive phenotypes.

## Figures and Tables

**Figure 1 cells-08-01118-f001:**
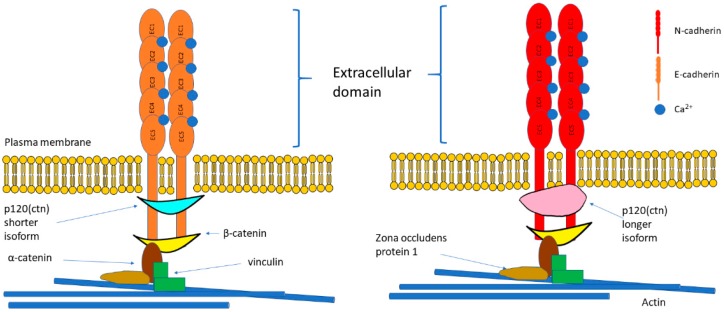
Structure of E-cadherin and N-cadherin. E-cadherin and N-cadherin are classical cadherins and share similar structures. They form cadherin-catenin complex where the cytoplasmic domain consists of EC repeats that bind with catenins to moderate the cytoskeletal filament containing actin. The structural difference between E-cadherin and N-cadherin is that E-cadherin binds with the shorter isoform of p120 catenin while N-cadherin binds with the longer isoform. Abbreviations: EC = extracellular cadherin; ctn: catenin.

**Figure 2 cells-08-01118-f002:**
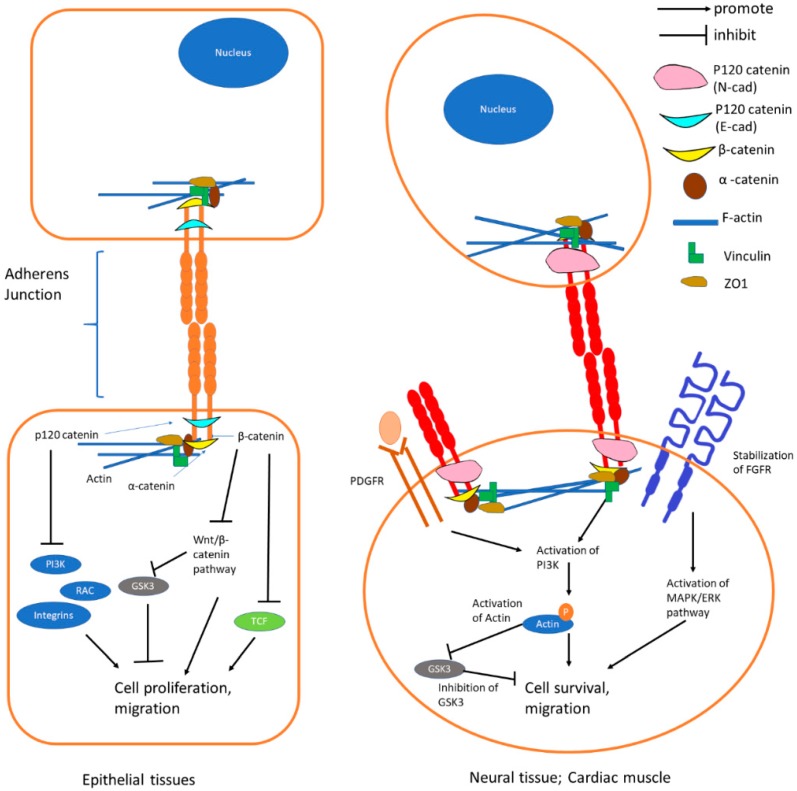
Functions of E-cadherin and N-cadherin. E-cadherin junctions form the stable adherens junction and enable strong cell–cell contact. As p120 catenin and β-catenin are strongly bound to the E-cadherin complex, they are not available to activate Wnt/β-catenin pathway and P13K pathway. N-cadherin junction enables the stabilization of fibroblast growth factor receptor (FGFR) which leads to activation of MAPK/ERK pathway and activates the PI3K pathway in association with PDGFR to enhance cell survival and migration. Abbreviations: PI3K = Phosphoinositide-3-kinase; RAC = Ras-related C3 Botulinum Toxin Substrate; GSK3 = Glycogen Synthase Kinase 3; TCF = T-cell Factor; PDGFR: Platelet-derived Growth Factor Receptor; FGFR: Fibroblast Growth Factor Receptor; MAPK = Mitogen-activated Protein Kinase; ERK = Extracellular Signal-regulated Kinases.

**Figure 3 cells-08-01118-f003:**
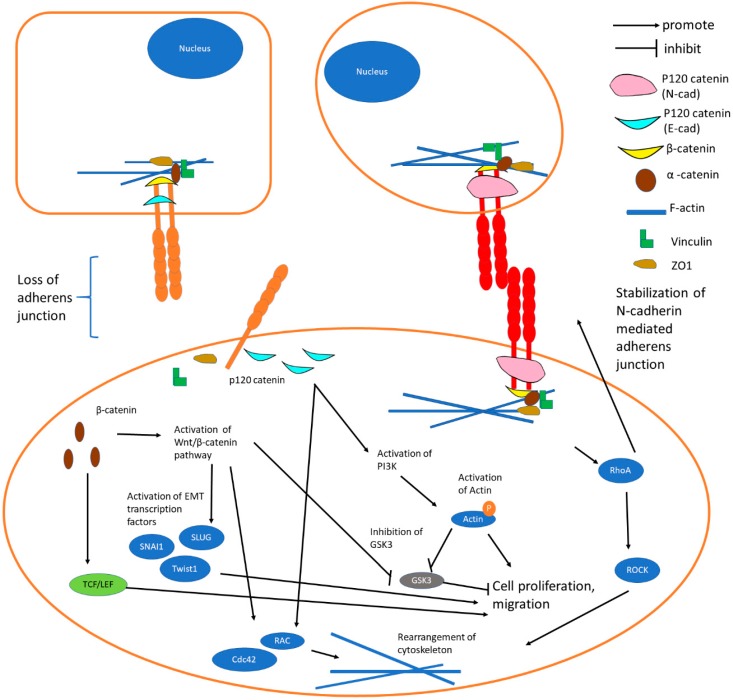
“Cadherin switching” is the downregulation of E-cadherins and upregulation of N-cadherins in EMT. E-cadherin-mediated adherens junctions disassociate due to the downregulation of E-cadherin while N-cadherin junctions establish a relatively weak adherens junction. Several signaling pathways such as Wnt/β-catenin, PI3K/AKT, TCF/lymphoid enhancer-binding factor (LEF) and RhoA will be activated by β-catenin following the loss of E-cadherin. Increased expression of RAC, cell division control protein 42 homolog (Cdc42), and RhoA leads to the rearrangement of cytoskeleton, whereby consequently the cell is changed from an adhesive state into motile state. Abbreviations: LEF = Lymphoid Enhancer-Binding Factor; RhoA = Ras Homolog Gene Family, member A; ROCK = Rho-associated, coiled-coil containing Protein Kinase; Cdc42 = Cell Division Control Protein 42 Homolog.

**Figure 4 cells-08-01118-f004:**
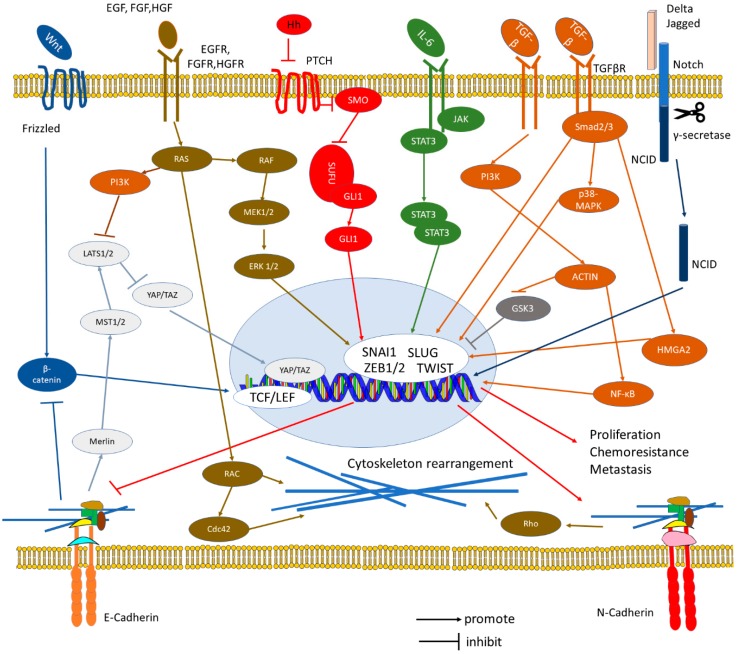
Signaling pathways involved in Epithelial-to-Mesenchymal Transition. Crosstalk between multiple signaling pathways increase the expression of EMT transcription factors including SNAI1, SLUG, TWIST, and ZEB leads to loss of epithelial characteristics and gain of mesenchymal characteristics. Wnt/β-catenin pathway induces EMT by interacting with TCF/LEF. Growth factors including Epidermal Growth Factor (EGF), FGF and hepatocyte growth factor (HGF) bind to their respective receptors to activate RAS/ERK signaling, RAS/PI3K signaling, and RAS/RAC signaling. Activation of RAS/RAC pathway subsequently leads to reorganization of cytoskeleton by elevating the expression of RAC and Cdc42. Hippo pathway inhibits YAP/TAZ pathway and EMT, but it will be suppressed when PI3K and F-actin are upregulated by other signaling pathways during EMT. Hedgehog (HH)/GLI and Janus kinase/signal transducer and activator of transcription (JAK/STAT) pathways increase SNAI1 expression and drive the EMT program. TGF-β pathway promotes EMT via either Smad or non-Smad signaling. TGF-β/Smad signaling activates Smad2/3 complex where the complex can interact with transcription factors in the nucleus to drive EMT, activates p38 MAPK pathway and increases the expression of HMGA2 to upregulate N-cadherin. Non-Smad signaling activates PI3K/AKT signaling to induce EMT by suppressing GSK3 and promoting the expression of NF-κB. NCID cleaved by activated Notch pathway will translocate into the nucleus to increase SLUG expression. N-cadherin upregulates the expression of RhoA to strengthen the cell–cell junction and enforce cytoskeleton rearrangement. Abbreviations: IL = Interleukin; LATS = Large Tumor Suppressor; NCID = Notch intracellular cytoplasmic domain; Rho = Ras homolog; PAK = Serine/threonine-protein Kinase; YAP = Yes-associated Protein; MEK = Mitogen-activated Protein Kinase; PTCH = Patched; SMO = Smoothened; SUFU = Suppressor of Fused Homolog; GLI 1 = Glioma-associated oncogene; JAK = Janus kinase; STAT = Signal Transducer and Activator of Transcription; NF-κB = Nuclear Factor Kappa-light-chain-enhancer of Activated B cells; HMGA2 = High mobility group A2; EGF: Epidermal Growth Factor.

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
