# Peer review of "The E-Cadherin and N-Cadherin Switch in Epithelial-to-Mesenchymal Transition: Signaling, Therapeutic Implications, and Challenges"

_cells, 2019, doi:10.3390/cells8101118_

Round 1

Reviewer 1 Report

Loh et al review the role of E-cadherin and N-Cadherin associated with epithelial-to-mesenchymal transition with respect to signaling processes as well as therapeutic implications and challenges. Downregulation of E-cadherin and concomitant upregulation of N-cadherin is a characteristic of EMT and therefore of great clinical interest. The authors summarize and assess the roles of E-cadherin and N-cadherin in invasion and metastasis of cancer cells, the involved signaling processes and the cross-talk with other EMT-regulating signaling pathways. Against this background, they discuss some natural compounds attributed with potential anti-EMT effect, such as curcumin, resveratrol, honokiol, epigallocatechin-3-gallate, S-allylcysteine and S-allyl mercaptocysteine, as well as ZL170.

This work is of interest to the readers of ‘Cells’. The review is well designed, and it provides data that support the authors’ conclusions.

Yet, some concerns need to be addressed.

Major comments to the authors:
1. The homophilic interaction of cadherins by domain exchange should at least be roughly described.
2. The cadherin-catenin complex does not directly bind to the actin cytoskeleton but via several adapter proteins, especially vinculin, which is highly extended N-cadherin containing adhesion complexes in contrast to E-cadherin containing adhesion complexes (Bertocchi et al. 2017: Nanoscale architecture of cadherin-based cell adhesions, 2017. PMID 27992406). Integrating mechanical and phosphorylation signals, it regulates the mechanical properties of cellular junctions. This should be noted in the text and also sketched in the figures.
3. A Pubmed search with the keywords ‘C-cadherin’, ‘N-cadherin’, ‘EMT’, ‘signaling’, ‘therapy’, and ‘challenges’ gives, among others, two publications that appear to be interesting in the context of this review: Wang S, et al. 2016: PBX3 promotes gastric cancer invasion and metastasis by inducing epithelial-mesenchymal transition. Oncol Lett. PMID 27900025; and Deng et al. 2017: The anti-tumor activities of Neferine on cell invasion and oxaliplatin sensitivity regulated by EMT via Snail signaling in hepatocellular carcinoma. Sci Rep. PMID 28134289).
4. To my knowledge the dispute is not yet settled, whether loss of E-cadherin is a major driver of EMT or merely consequential to EMT. A corresponding discussion would be desirable. See, e.g., Wong et al. 2017: E-cadherin: Its dysregulation in carcinogenesis and clinical implications. Crit Rev Oncol Hematol. PMID 29279096.
5. In the context of this review, soluble cadherins, e.g., as tumor markers, would be interesting: In a phase I clinical trial, soluble E-cadherin has been identified as a biomarker to monitor cancer therapy (Reckamp et al. 2008: Tumor response to combination celecoxib and erlotinib therapy in non-small cell lung cancer is associated with a low baseline matrix metalloproteinase-9 and a decline in serum-soluble E-cadherin. J Thorac Oncol. PMID 18303430.
6. Line 215: Neuropilin-2 is discussed as a regulator of E-cadherin, here. To my knowledge, neuropilin-1 upregulates N-cadherin, which is not mentioned, but could be of interest in the context of this review (cf., Zhou et al. 2016: A novel association of neuropilin-1 and MUC1 in pancreatic ductal adenocarcinoma: role in induction of VEGF signaling and angiogenesis. Oncogene. PMID 26804176).
7. Line 296ff: Another mechanism involved in cancer initiation that needs to be mentioned here is a disturbed sequestering of beta-catenin (Pecina-Slaus et al. 2003: Tumor suppressor gene E-cadherin and its role in normal and malignant cells. Cancer Cell Int. PMID 14613514).
8. Why is luteolin not discussed in this review? Among other effects, it induces E-cadherin and downregulates N-cadherin, and it reverses EMT (cf., Imran et al 2019: Luteolin, a flavonoid, as an anticancer agent: A review. Biomed Pharmacother. PMID 30798142).
9. Why is kaempferol not discusses in this review? It suppresses expression of N-cadherin, E-cadherin and other EMT-related proteins. (cf., Imran et al 2018: Chemo‐preventive and therapeutic effect of the dietary flavonoid kaempferol: A comprehensive review. Phytother Res, 30402931).
10. References should be carefully checked:
• line 389: Reference 115 is missing
• The abbreviated first names of some last authors are fused with single-letter-abbreviated journal titles (correspondingly, journal names are missing in these references), e.g. references 7, 10-12, 53.
• Some authors are listed with incorrect abbreviated first names, e.g., reference 8: ‘Kolligs FT’ instead of ‘Kolligs FTJB’.
• Some Journal titles are missing, e.g., reference 8: ‘Bioessays’.
• Some Journal titles are wrong, e.g., reference 97: ‘Biochem Soc Trans’ instead of ‘Portland Press Limited’ (here also issue and page numbers are missing, too)
• Some publication years are missing, e.g., reference 14.
• Some page numbers are incomplete, e.g., reference 42, 43, 50, 51, 105.
• References should be formatted consistently (type and order of record entries, abbreviations, capitalization, page numbers).

Minor comments to the authors:
1. Line 37: In the introduction, the discoverers of cadherin and their functions should not go unmentioned.
2. Line 39/40: The size of cadherins in KDa would be appreciated.
3. Figure 1: As differential binding to short/long p120 isoform is an important information, this may be highlighted in the figure by a bright color and a full vs. a truncated shape.
4. Line 53: in the figure, ‘EC’ does not occur but, nearly unreadably small, ‘EC1’ through ‘EC5’. They should be labeled as ‘extracellular domains’ rather than extracellular cadherins’.
5. Figures 2, 3: A legend explaining the symbols for F-actin, catenins, and p120s would be helpful.
6. Figure 3: Why is the (F-actin?) cytoskeleton blue (top) or red (bottom)? Rearrangement of the fibers could be more clearly represented by a different geometric arrangement of the lines.
7. Line 111: Cadherins do not switch. Their expression levels (and/or subcellular localization?) change.
8. Line 140: I do not understand, why the extrinsic apoptosis pathway is mentioned in the context of EMT with a reference, but the intrinsic pathway not.
9. Line 150: The logical order in this sentence seems wrong to me.
10. Figure 4: Why is the plasma membrane interrupted, e.g., at the bottom in in the middle?
11. Abbreviations should be explained at their first use (e.g., HH: in line 163 instead of line 253)
12. Line 172: Rho is most likely not rhodopsin as stated here.
13. Line 181/182: This sentence seems to be incomplete.
14. Line 216: The abbreviation ETV4 is not explained. A list of all abbreviations would be helpful.
15. Line 224: What are ‘null embryonal stem cells’? Also, there is no reference given for the statements of the previous and this sentence.
16. Line 259-261: The reference for this finding (#86) is missing here, while it is given at the end of the paragraph.
17. Line 326-332. In the entire paragraph references are missing.
18. Line 337: No reference is specified.
19. Line 341-342: Does this statement belong to reference #94?
20. Line 358-360: Does this statement belong to reference #106?
21. Line 376: No reference is specified.
22. Line 382: What is meant with ‘perturbations in actin’: altered expression, subcellular distribution, polymerization, structure?
23. Lin 389: Reference #115 is not listed under ‘References’.
24. Line 449: What is meant with ‘to abrogate cancer associated fibroblast’? Does the tumor stroma not contain this cell type any more after administration of curcumin?
25. Lines 462-469: No reference is specified.
26. Line 476-478: No reference is specified.
27. Line 484-486: No reference is specified.
28. Line 500: Please specify: Did resveratrol inhibit proliferation, migration or invasion of carcinoma cells?
29. Line 514: It may be helpful for the reader to know that adriamycin increases E-cadherin expression in a human breast cancer cell line (Yang et al. 1999: Adriamycin activates E-cadherin-mediated cell-cell adhesion in human breast cancer cells. Int J Oncol. PMID 10568816).
30. Line 514-515: No reference for resveratrol as sensitizer to doxorubicin and promoter of apoptosis is specified.
31. Line 525: The word ‘they’ incorrectly implies that references #88 and #160 have common authors.
32. Line 565: Different abbreviations, such as, for example, Snail (line 565), SNAII (e.g., line 121), SNAI1 (line 156) or HH (line 163) and Hh (table 1) for a protein should be avoided.
33. Line 576-577: No reference is specified. It should be: Wang X, et al. 2011: Honokiol crosses BBB and BCSFB, and inhibits brain tumor growth in rat 9L intracerebral gliosarcoma model and human U251 xenograft glioma model. PLoS One. PMID 21559510
34. Line 596-597: No reference is specified.
35. Line 607-612: ‘some natural compounds’ could be specified and references could be given for the ‘success stories of paclitaxel and vincristine’.
36. Line 616-619: No reference is specified.
37. Line 617: Together with bioavailability, pharmacokinetics and pharmacodynamics of promising compounds could be discussed, too.
38. Line 621-629: No reference is specified.
39. Line 631-636: No reference is specified. At the very least, a review should be provided to help interested readers get started with the literature on nanotechnology therapeutic approaches.

Author Response

Dear Reviewer,

Thank you for your valuable comments and suggestions. Please see the attachment for your review. Our responses were in bold. 

Reviewer 2 Report

The review entitled "The E-cadherin and N-cadherin Associated with Epithelial to Mesenchymal Transition: Signaling, Therapeutic Implications and Challenges" seems to have some value in reviewing the impact of the cadherin molecules and EMT progression, but is poorly organized. That the title seems to be missing a word either before or after "Associated" was a clue to the lack of organization. In many cases, the paragraphs are a collection of facts with no apparent thesis. Specifically, the paragraph beginning with line 120 compared with the following paragraph. The paragraph labeled "4.1 Inhibitors" is a jumble of facts. I was expecting a preview of molecules to be discussed below. The entire section on resveratrol treatment impacts on EMT is a jumble. For instance, the references to the important mir-200 effects are in two separate paragraphs when I believe these belong together.

Other things:

I think elements of the figures should be referenced within the text, not at the end of the preceding section.

I think line 54 should reading "founding member"?

In line 292, the verb conjugation should be "decreased"/

In line 359, the phrase "and not E-cadherin itself" should be offset with commas.

In line 376, the first sentence has 2 verbs.

Reference 115 is not present in the bibliography.

There seems to be a typo in the title of reference 46? FAFalphac?

In line 416, there should be a hyphen after "E" preceding "and".

Finally, I am not sure why I need Table 1.

Author Response

Dear Reviewer,

Thank you for your valuable comments and suggestions. Please see the attachment for your review. Our responses are in bold. 

Reviewer 3 Report

The authors have reviewed the role of E-cad/N-cad associated with EMT in this manuscript. A worthwhile topic to review, however, the manuscript is filled with examples of not following the best reference practices and thus potentially misleading at many places. Many references are not correctly cited (i.e. the claim made in the review for that paper is not really mentioned in that paper), many of them are missing (i.e. a claim is being made but without identifying any reference there), and many important references about a topic have not been discussed at all. Put together, these limitations severely restrict the usefulness of this review article in the field, and thus needs a full overhaul. Please see my detailed comments below:

The authors claim that “cancer cells commonly use aerobic glycolysis rather than mitochondrial OXPHOS to acquire ATP as energy to sustain their high rates of proliferation [22,23]”. There are ample reports suggesting the opposite as well as few reports suggesting that cancer cells may use a mixture of both glycolysis and OXPHOS mechanisms for their metabolic reprogramming; for instance, see Jia et al. PNAS 2019, Yu et al. Cancer Res 2017

The authors mention that “The adherens junctions are comprised of cadherins that are correlated to 104 the actin cytoskeleton via binding to α-, β- and γ-catenins [37].”, but the stated reference is about Wnt signaling predominantly.

The authors state that “However, N-cadherin can inhibit the EMT and establish strong adherens junctions in the neural tissue”. No reference is cited for this claim.

The authors state that “GLI-inhibitor (Gli-i) downregulated E-cadherin while 272 upregulated N-cadherin and β-catenin”. However, the stated reference (#90) actually says the exact opposite.

The authors state that “also stated that non-canonical GLI1 activation of the HH pathway and MAPK pathway could lead to decrease of E-cadherin expression in thyroid cancer cell lines, which subsequently caused by BRAF mutation-induced SNAIl expression [85].” Ref. 85 does not state the abovementioned statements.

The authors state that “Casticin or SMO-inhibitor (cyclopamine) treated ovarian cancer cell line showed notably increased in E-cadherin expression and reduced in N-cadherin expression”. This aspect is not stated in the reference claimed.

The authors also portray EMT as a binary picture with a “cadherin switch”. There are ample studies now showing that EMT is not a binary phenomenon; instead, cells can undergo one or the other manifestations of partial EMT (Jolly et al. Mol Oncol 2017).

Author Response

(The authors gave the same response as above.)

Round 2

Reviewer 1 Report

All criticisms and comments have been taken into account and the work is suitable for publication in the current version.

However, attention should be paid to the following:

In lines 117 and 119, “casual” seems to be a misspelling of “causal”. In figures 2 and 3, a symbol for vinculin has been added without being explained in the legend. In line 583, Linn is not part of the Latin name of the plant but the abbreviation of Carl Linnaeus who first described it. As such, it should not be italicized and as usually abbreviated as “L.” rather than “Linn”. Other than stated in the authors' reply to #10 of “major comments to the authors”, the formatting of the references is not yet uniform, e.g., reference 1 (and others): journal title with only first word capitalized vs. reference 2 (and others): all words capitalized. doi: sometimes stated (reference 2 and others), sometimes not (reference 1 and others); in reference 90 underlined references 2 and 100 versus all others: throughout in capital letters reference 1 (and others): full journal title versus reference 9 (and others): abbreviated journal title reference 15: journal title missing, journal volume missing, and page numbers missing. Instead, the publisher is specified.

Author Response

We revised these in the paper.

Reviewer 2 Report

This edition of the manuscript is much improved. A few minor edits are necessary:

Beginning on line 116, "that decreased by E-cadherin expression" needs a verb change. In the same paragraph as #1, the word "casual" is used twice and I believe is should be "causal". In line 565, add a hyphen between Twist dependent". In line 669, add a hyphen after "EGF" and "TGF-b" to indicate each are "induced". In line 780 mdm2 should be in all capitals since it refers to the human gene name.

Author Response

We revised these in the paper.

Reviewer 3 Report

The authors have addressed the scientific queries I had; the quality of the language and professional editing can definitely be improved.

We revised these in the paper.